# DnaK supports intracellular persistence of *Staphylococcus xylosus* and confers mechanical resilience to a human breast cancer cell line

**Lei Ye, Guozheng Yu, Yi Cheng, Lijuan Fan**◉*

Department of Thyroid and Breast Surgery, Huangshi Central Hospital, Affiliated Hospital of Hubei Polytechnic University, Huangshi, China

* 18672014609@163.com

## Abstract

Intratumoral *Staphylococcus xylosus* enhances the ability of breast cancer cells to survive mechanical shear stress, a critical barrier encountered during hematogenous metastasis. However, the bacterial determinants underlying this effect remain unclear. Here, we identify the bacterial molecular chaperone DnaK as a key factor enabling *S. xylosus* to promote shear-stress tolerance in a human breast cancer cell line. Deletion of *dnaK* did not affect bacterial adhesion to or invasion of MDA-MB-231 cells but significantly reduced sustained intracellular survival. Under oxidative and acidic stress conditions, the Δ*dnaK* mutant showed reduced survival compared with the wild-type strain, and its ability to enhance tumor-cell viability under shear stress was markedly impaired. Using a breast cancer–on–a–chip microfluidic model, we demonstrate that infection with wild-type or complemented *Staphylococcus xylosus* confers increased tumor-cell viability under laminar shear stress in a time-dependent manner, whereas cells infected with the Δ*dnaK* mutant fail to acquire shear-stress resistance and resemble uninfected controls. Together, these findings establish DnaK-dependent intracellular persistence of *S. xylosus* as a critical determinant of tumor-cell survival under mechanical stress, linking a conserved bacterial stress-response protein to cancer cell biomechanics in a metastasis-relevant context.

## Introduction

Cancer remains a leading cause of morbidity and mortality worldwide, accounting for nearly 10 million deaths annually [1]. In addition to genetic and environmental factors, emerging studies have highlighted the presence and functional significance of microbiota within tumor tissues [2–4]. Far from being passive bystanders, these intratumoral microbes have been shown to shape local immune responses [5], modulate chemotherapeutic efficacy [6], and influence tumor progression and metastasis [7]. In breast cancer specifically, bacterial taxa such as *Streptococcus*, *Lactobacillus*, and

**Data availability statement:** All relevant data are within the paper.

**Funding:** The author(s) received no specific funding for this work.

**Competing interests:** The authors have declared that no competing interests exist.

*Staphylococcus* have been recurrently identified in both primary tumors and metastatic lesions [7]. Their capacity to reside intracellularly within tumor and immune cells suggests that these microbes may actively adapt to and exploit the unique stressors of the tumor microenvironment.

Among the genera identified in breast tumors, *Staphylococcus xylosus* has garnered attention due to its ability to survive intracellularly and its potential to modulate tumor cell behavior. While traditionally regarded as a commensal organism found on the skin and mucous membranes [8], *S. xylosus* can act as an opportunistic pathogen under certain conditions, including nosocomial infections and dysbiosis-associated inflammation [9,10]. Recent animal studies have demonstrated that *S. xylosus* introduced into mammary tumors can persist within tumor cells and enhance their resistance to mechanical stresses such as fluid shear, a critical factor during hematogenous metastasis [11]. Despite this, the bacterial determinants that enable *S. xylosus* to persist in the tumor niche and support cancer cell survival under adverse conditions remain poorly characterized.

One candidate of interest is DnaK, a highly conserved molecular chaperone belonging to the Hsp70 family [12,13]. In bacteria, DnaK is central to proteostasis, working in concert with co-chaperones DnaJ and GrpE to refold misfolded proteins and prevent aggregation during environmental stress [14]. Its role extends beyond general stress tolerance: DnaK has been implicated in the virulence of diverse pathogens, including *Staphylococcus aureus* [15], *Salmonella enterica* [16,17], *Helicobacter pylori* [18], and *Mycobacterium tuberculosis* [19]. In these organisms, loss of DnaK compromises survival under oxidative stress, impairs intracellular persistence, and attenuates pathogenicity. These findings suggest that DnaK is a key facilitator of bacterial adaptation to host-imposed stresses, including reactive oxygen species (ROS), low pH, and temperature fluctuations.

Notably, the functional role of DnaK in coagulase-negative staphylococci such as *S. xylosus* has yet to be systematically studied, particularly in the context of tumor colonization. Given the increasingly recognized impact of intratumoral bacteria on cancer cell physiology and disease progression, elucidating how *S. xylosus* survives and persists within tumor cells is of both microbiological and oncological relevance. In this study, we sought to investigate the contribution of DnaK to the physiological fitness of *S. xylosus* and its potential role in mediating stress adaptation within the tumor microenvironment. Understanding the mechanisms by which bacteria cope with intracellular stressors may provide new insights into microbe–tumor interactions and identify microbial targets relevant to cancer progression and therapy.

## Materials and methods

### Bacterial strains and culture conditions

The bacterial strains and plasmids used in this study are listed in Table 1. *Staphylococcus xylosus* wild-type strain, originally isolated from a human breast tumor biopsy, served as the parental background for genetic manipulation [11]. An in-frame deletion mutant of the *dnaK* gene (Δ*dnaK*) and its genetically complemented derivative

**Table 1. Bacterial strains and plasmids used in this study.**

| Strain/ Plasmid | Description | Source/ Reference |
|---|---|---|
| *S. xylosus* WT | Wild-type clinical isolate from human breast tumor | Provided by [11] |
| Δ*dnaK* | In-frame deletion of *dnaK* in WT background | This study |
| Δ*dnaK*-C | Δ*dnaK* strain complemented with *dnaK* under native promoter on pLOW | This study |
| pIMAY | Temperature-sensitive allelic exchange vector | [39] |
| pLOW | Low-copy shuttle vector for *Staphylococcus* spp. | [40] |

(Δ*dnaK*-C) were constructed as described below. Strains were routinely cultured in tryptic soy broth (TSB; BD Difco) at 37 °C with shaking at 200 rpm. When appropriate, erythromycin (5 μg/mL) or chloramphenicol (10 μg/mL) was added for plasmid selection.

## Construction of the Δ*dnaK* mutant and complemented strain

An in-frame deletion of *dnaK* was generated using a temperature-sensitive allelic exchange system based on the pIMAY plasmid. Briefly, ~1,000-bp upstream and downstream flanking regions of the *dnaK* open reading frame were amplified from genomic DNA using primers listed in Table 2. The PCR fragments were cloned into the EcoRI/HindIII sites of pIMAY, and the resulting construct was introduced into *S. xylosus* via electroporation. Mutants resulting from double-crossover recombination were selected by temperature shift and counterselection on anhydrotetracycline-free medium, and deletion was confirmed by colony PCR and Sanger sequencing. For genetic complementation, the full-length *dnaK* gene including its native promoter (~250 bp upstream) was amplified and cloned into the low-copy shuttle vector pLOW. The resulting plasmid was introduced into the Δ*dnaK* mutant by electroporation, yielding the complemented strain Δ*dnaK*-C. All constructs were verified by and sequencing.

## Growth characterization of bacterial strains

Growth characterization was performed to compare the in vitro proliferation of *Staphylococcus xylosus* wild-type, Δ*dnaK*, and Δ*dnaK-C* strains under standard culture conditions. Each strain was inoculated from overnight cultures into fresh

**Table 2. Primers used for *dnaK* deletion and complementation.**

| Primer Name | Sequence (5'→3') | Purpose |
|---|---|---|
| *dnaK*-Up-F | **GAATTC**GGCATTTTTATGGGCGGAAAGG | Amplify upstream region of *dnaK* (EcoRI) |
| *dnaK*-Up-R | **GGTACC**ACTCATAAATAAATTCCTCCTGT | Upstream reverse (KpnI) |
| *dnaK*-Down-F | **GGTACC**TAGGTGTAGTCATGGACTAATT | Downstream forward (KpnI) |
| *dnaK*-Down-R | **AAGCTT**CGTTGCTTGGGCAATGCTAAC | Amplify downstream region of *dnaK* (HindIII) |
| *dnaK*-comp-F | **GAATTC**ACAGTTTAAATCGCTTAAAAAAGG | Amplify full-length *dnaK* with promoter (EcoRI) |
| *dnaK*-comp-R | **AAGCTT**AATAATCTCTTTTGGCCACTG | Complementation (HindIII) |
| *dnaK*-check-F | CCAGACAATGAAAGAATATAAAG | PCR confirmation of Δ*dnaK* |
| *dnaK*-check-R | CTTTAAACTTCTCGTCAGCACCT | PCR confirmation of Δ*dnaK* |

Note: Restriction enzyme recognition sites at both ends of the primers are indicated in **bold** and are used for restriction-ligation cloning.

tryptic soy broth (TSB; BD Difco) at an initial optical density ($OD_{600}$) of 0.05. Cultures were incubated at 37 °C with shaking at 200 rpm. Growth was monitored by measuring $OD_{600}$ every hour using a BioTek Synergy HTX microplate reader. In parallel, aliquots were collected at designated time points for enumeration of viable bacteria by plating serial dilutions on TSB agar to determine colony-forming units (CFU/mL). All assays were performed in triplicate with three independent biological replicates.

## Mammalian cell culture and infection assays

The human breast cancer cell line MDA-MB-231 was obtained from ATCC and cultured in DMEM supplemented with 10% fetal bovine serum (Gibco), 1% penicillin-streptomycin, and 2 mM L-glutamine. Cells were maintained at 37°C in a humidified atmosphere with 5% $CO_2$. For infection assays, antibiotics were removed 24 hours prior. Cells were seeded at $5 \times 10^5$ per well in 12-well plates. Bacteria were grown to mid-log phase ($OD_{600} \approx 0.6$), washed, and resuspended in serum-free DMEM. Cells were infected at MOI = 100:1 and incubated for 2 h at 37°C. Following infection, cells were washed three times with PBS and incubated with medium containing gentamicin (100 µg/mL) for 1 h to kill extracellular bacteria. Cells were then washed and maintained in medium with 10 µg/mL gentamicin for the remainder of the assay. Intracellular CFUs were enumerated at indicated time points by lysing cells with 0.1% Triton X-100 and plating serial dilutions on TSB agar.

## Adhesion and invasion assays

Adhesion and invasion assays were performed as previously described with minor modifications [20]. Breast cancer cells (MDA-MB-231) were seeded into 24-well plates at $5 \times 10^5$ cells/well and cultured overnight in DMEM supplemented with 10% fetal bovine serum. Mid-logarithmic phase *S. xylosus* wild-type, Δ*dnaK*, and Δ*dnaK-C* strains were washed twice with PBS and added to host cells at a multiplicity of infection (MOI) of 50:1. After incubation for 2 h at 37 °C with 5% $CO_2$, wells were washed three times with PBS to remove non-adherent bacteria. For the adhesion assay, cells were lysed immediately with 0.1% Triton X-100 in PBS, and adherent bacteria were enumerated by plating serial dilutions on TSB agar. For the invasion assay, following the 2-h infection period, cells were incubated with DMEM containing gentamicin (100 µg/mL) for 1 h to kill extracellular bacteria. After three washes with PBS, cells were lysed, and intracellular bacteria were quantified by CFU enumeration. *Salmonella enterica* serovar Typhimurium and *Escherichia coli* DH5α were used as positive and negative controls, respectively. Each experiment was performed in triplicate and repeated at least three times independently.

## Quantification of total bacterial load by species-specific qPCR

Total bacterial load per well was determined by quantitative real-time PCR (qPCR) using *Staphylococcus xylosus*–specific primers targeting the 16S rRNA gene. After infection and washing to remove non-adherent bacteria, host cells and associated bacteria were lysed in 200 µL of lysis buffer (10 mM Tris-HCl [pH 8.0], 1 mM EDTA, 0.5% Triton X-100) followed by heating at 95 °C for 10 min. Lysates were centrifuged (10,000 × g, 5 min) and 2 µL of the supernatant was used as template DNA. qPCR reactions (20 µL total volume) were prepared with SYBR Green Master Mix (Takara) and primers Sx-16S-F (5′-AGTGGCGGACGGGTGAGTA-3′) and Sx-16S-R (5′-GACGACCATCGTTTACGGCG-3′). Amplification was performed on a QuantStudio 5 Real-Time PCR System (Applied Biosystems) with the following cycling conditions: 95 °C for 3 min, followed by 40 cycles of 95 °C for 10 s and 60 °C for 30 s. A standard curve was generated from serial dilutions of genomic DNA extracted from a known number of *S. xylosus* cells to convert Ct values to bacterial cell equivalents.

## Stress resistance assays

To evaluate bacterial tolerance to stress conditions, overnight cultures were diluted 1:100 into fresh TSB and grown to mid-log phase. For acid stress, bacterial suspensions were adjusted to pH 5.5 with HCl and incubated for 1 h. For

oxidative stress, cultures were exposed to 5 mM hydrogen peroxide ($H_2O_2$) for 30 min at 37°C. To test hypoxia tolerance, bacteria were cultured in an anaerobic chamber (1% $O_2$, 5% $CO_2$, balanced $N_2$; Whitley H35 Hypoxystation) for 12 h. LL-37 peptide (InvivoGen) was used at 10 μg/mL for 1 h in PBS with 1% BSA. Viability was determined by CFU plating.

### Microfluidic shear stress assay

A breast cancer–on–a–chip microfluidic system was employed to model fluid shear stress. MDA-MB-231 cells were seeded into polydimethylsiloxane (PDMS)-based microfluidic channels (channel height 100 μm, width 1 mm) bonded to glass slides using plasma treatment. After cell monolayers formed (~48 h), cells were infected with *S. xylosus* strains as described above.

Following infection and antibiotic treatment, the microfluidic device was connected to a programmable syringe pump (Harvard Apparatus) to generate laminar flow (0.2 dyn/cm²) for 8 h at 37°C. Cell viability was assessed using LIVE/DEAD Viability/Cytotoxicity Kit (Thermo Fisher) and quantified by fluorescence microscopy. Three independent devices were used per strain per experiment.

### Statistical analysis

All experiments were performed with at least three independent biological replicates. Quantitative data are presented as mean ± standard error of the mean (SEM). Statistical analyses were carried out using GraphPad Prism version 9.0 (Graph-Pad Software, San Diego, CA, USA). One-way analysis of variance (ANOVA) followed by Tukey's multiple-comparison test was used for comparisons among three or more groups (Figs 1B, 2C, 3, 4 and 5B). Student's *t*-test was used for pairwise comparisons where appropriate. A *p* value < 0.05 was considered statistically significant.

## Results

### Deletion of *dnaK* does not impair bacterial growth in vitro

To determine the physiological impact of *dnaK* disruption, an in-frame Δ*dnaK* mutant was constructed in *Staphylococcus xylosus* (Fig 1A), along with a complemented strain (Δ*dnaK*-C) harboring a plasmid-expressed copy of *dnaK*. Under standard aerobic culture conditions, the Δ*dnaK* strain exhibited growth dynamics comparable to both the wild-type and complemented strains, with no significant differences observed in lag phase, exponential growth rate, or maximum cell density (Fig 1B). These findings indicate that *dnaK* deletion does not affect bacterial proliferation in nutrient-rich environments, and that the genetic manipulation does not introduce growth defects independent of stress response.

### DnaK is essential for intracellular persistence but not for adhesion or invasion

The ability of wild-type, ΔdnaK, and ΔdnaK-C strains to interact with breast cancer cells was assessed using an in vitro infection model. Bacterial adhesion and invasion were quantified at 2 h post-infection, and all strains exhibited comparable levels of host-cell association and intracellular entry at this early time point (Fig 2A and 2B).To distinguish initial entry from intracellular persistence, total bacterial burden per well was next quantified at 24 h post-infection using *S. xylosus*–specific qPCR, which revealed no significant differences among the three strains, indicating equivalent initial infection levels and bacterial association with host cells (Fig 2C). In contrast, intracellular survival was assessed at 24 h post-infection following gentamicin protection and host-cell lysis, and enumeration of viable intracellular CFUs demonstrated that the ΔdnaK mutant exhibited a more than three-fold reduction in intracellular survival compared with the wild-type strain (Fig 2D). This defect was fully restored in the complemented strain. Together, these results demonstrate that DnaK is required for long-term intracellular persistence at later post-infection time points, but is dispensable for host-cell entry at early stages of infection.

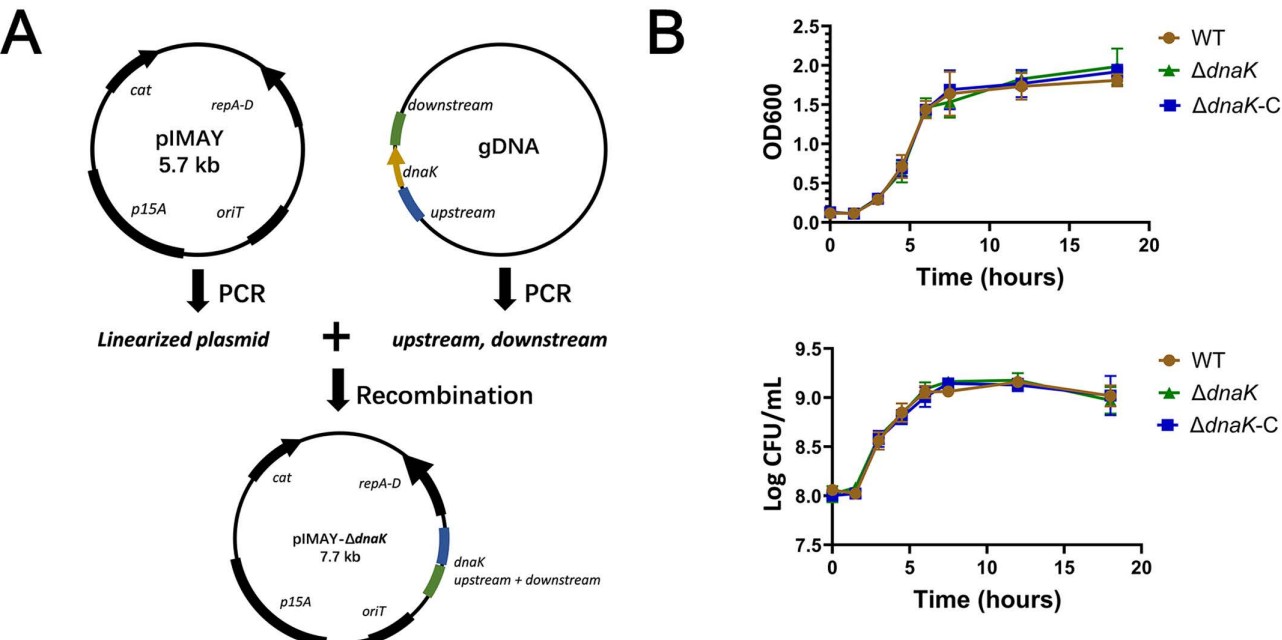

**Fig 1. Construction of the Δ*dnaK* mutant and assessment of bacterial growth under aerobic conditions. (A)** Schematic representation of the in-frame deletion strategy used to generate the ΔdnaK mutant of *Staphylococcus xylosus*. Approximately 1-kb regions flanking the *dnaK* coding sequence were cloned into the temperature-sensitive shuttle vector pIMAY for allelic exchange. **(B)** Growth kinetics of wild-type, Δ*dnaK*, and Δ*dnaK*-C strains cultured in TSB at 37 °C with shaking. OD$_{600}$ measurements and CFU enumeration were performed at indicated time points. Data represent means ± SEM from three independent experiments. Statistical comparisons among groups were analyzed by one-way ANOVA followed by Tukey's post hoc test. ns, not significant.

## DnaK is not required for survival under hypoxia or exposure to antimicrobial peptides

To define the stress specificity of DnaK function, bacterial survival was tested under hypoxic and antimicrobial peptide stress. Under 1% oxygen, all three strains—wild-type, Δ*dnaK*, and Δ*dnaK*-C—displayed equivalent growth kinetics, indicating that DnaK is not required for adaptation to hypoxia (Fig 3A). Similarly, exposure to LL-37, a host-derived antimicrobial peptide, resulted in no significant difference in bacterial viability across strains (Fig 3B). These data demonstrate that DnaK is dispensable for resistance to hypoxic and antimicrobial peptide stress.

## DnaK promotes adaptation to oxidative and acid stress

Bacterial survival under oxidative and acidic conditions was next assessed. The Δ*dnaK* mutant exhibited significantly increased sensitivity to hydrogen peroxide compared to the wild-type and Δ*dnaK*-C strains (Fig 4A). Likewise, under acidic conditions (pH 5.5), the mutant strain showed reduced viability, while both the wild-type and complemented strains remained resistant (Fig 4B). These findings indicate that DnaK selectively promotes survival under oxidative and acid stress, and that complementation fully restores this function, confirming the phenotype is attributable to *dnaK* deletion.

## DnaK enables *S. xylosus*-infected tumor cells to resist mechanical stress in a microfluidic model

To examine the impact of DnaK-mediated bacterial persistence on tumor-cell survival under mechanical stress, MDA-MB-231 cells were infected with wild-type, ΔdnaK, or ΔdnaK-C Staphylococcus xylosus and subjected to laminar shear stress using a breast cancer–on–a–chip microfluidic platform (Fig 5A). Time-course analysis revealed

 

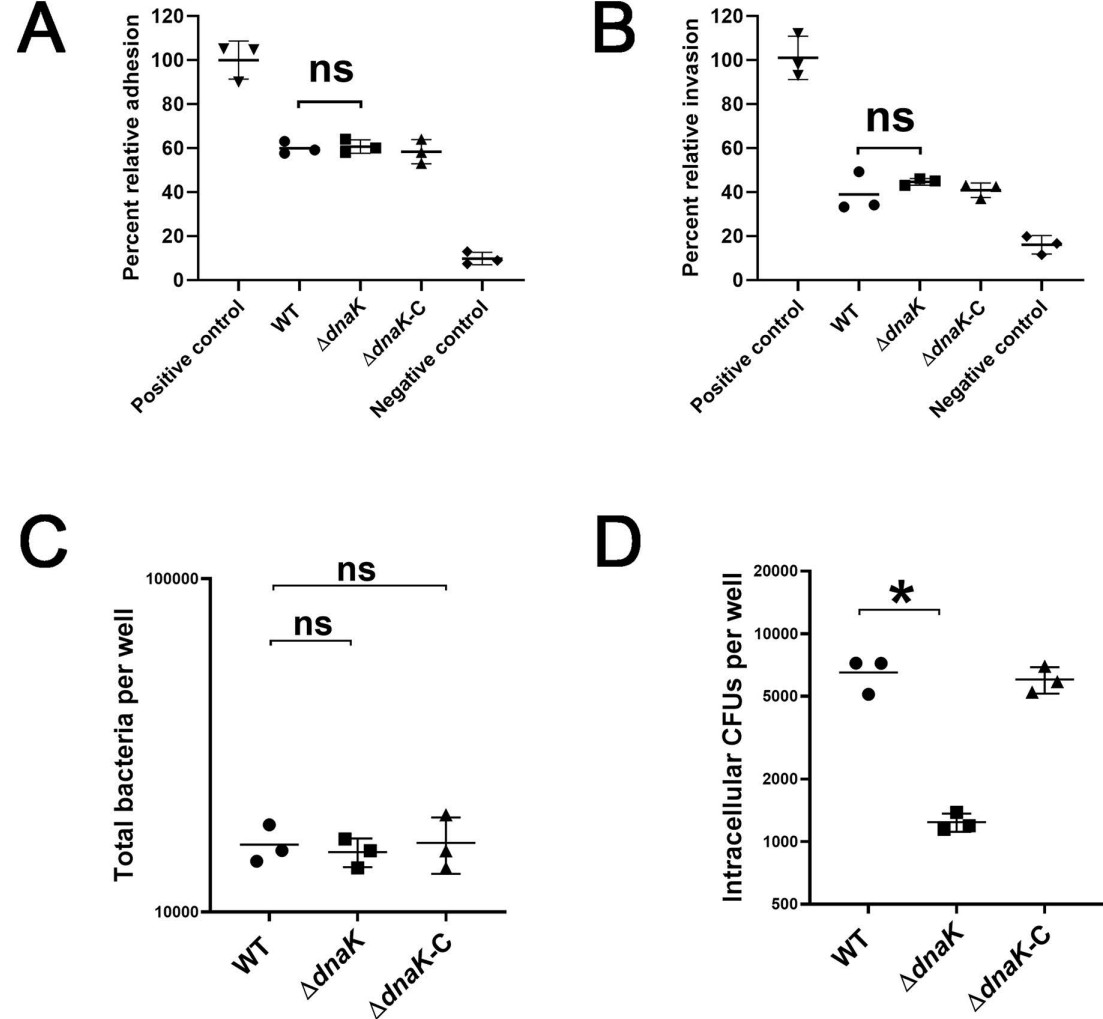

**Fig 2. Assessment of host cell adhesion, invasion, and intracellular survival. (A)** Quantification of bacterial adhesion to breast cancer cells at 2 h post-infection. **(B)** Quantification of bacterial invasion into breast cancer cells at 2 h post-infection. *Salmonella enterica* serovar Typhimurium was used as a positive control for adhesion and invasion, while *E. coli* DH5α served as a negative control. For both adhesion and invasion assays, *Salmonella enterica* serovar Typhimurium was used as a positive control, and *Escherichia coli* DH5α served as a negative control. **(C)** Quantification of total bacterial load per well at 24 h post-infection, determined by *S. xylosus*-specific qPCR. **(D)** Enumeration of intracellular CFUs of wild-type, Δ*dnaK*, and Δ*dnaK*-C strains at 24 h post-infection following gentamicin treatment and host-cell lysis. Data represent means ± SEM from three independent experiments performed in triplicate. Statistical significance was determined by one-way ANOVA followed by Tukey's multiple-comparison test ($p < 0.05$; ns, not significant).

that cells infected with wild-type or ΔdnaK-C strains maintained higher viability during shear-stress exposure compared with uninfected cells, whereas ΔdnaK-infected cells showed reduced survival that closely resembled uninfected controls (Fig 5B). Consistent with these observations, direct comparison at the 4 h time point demonstrated significantly increased viability in cells infected with wild-type or complemented strains relative to both uninfected and ΔdnaK-infected cells (Fig 5C). Together, these data indicate that DnaK is required for S. xylosus–associated enhancement of tumor-cell survival under mechanical stress, and that this phenotype is fully restored by genetic complementation.

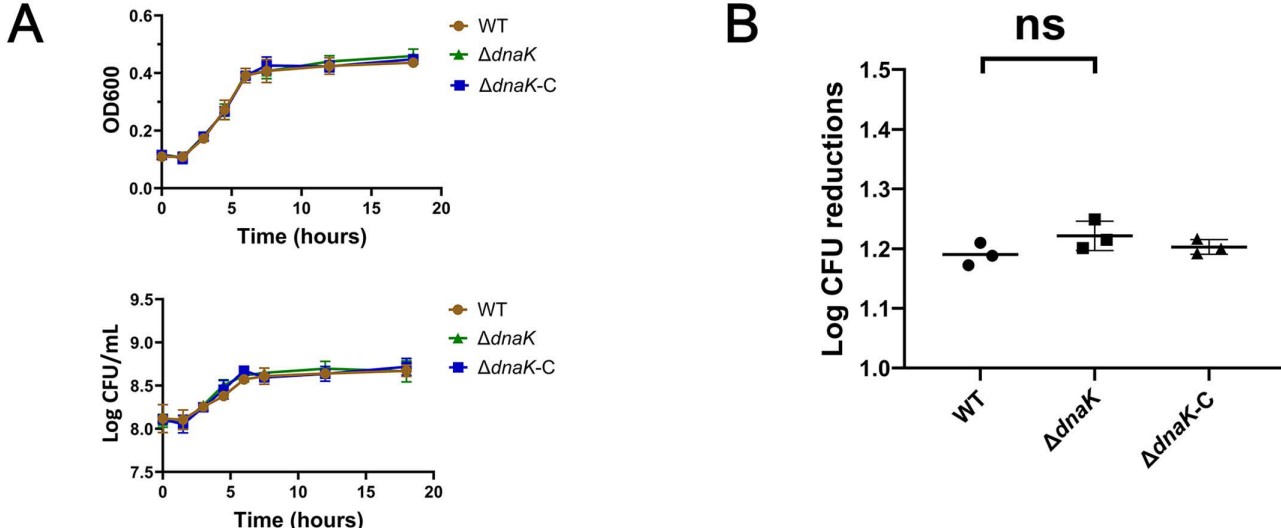

**Fig 3. DnaK is dispensable for resistance to hypoxic stress and antimicrobial peptide exposure. (A)** Growth of wild-type, ΔdnaK, and ΔdnaK-C strains under hypoxic conditions (1% $O_2$). $OD_{600}$ measurements and CFU enumeration were performed at indicated time points. **(B)** Survival of strains following treatment with LL-37 (10 µg/mL, 2 h). Viable CFUs were determined by plating. Data represent means±SEM from three independent experiments. Differences among groups were analyzed by one-way ANOVA. $p < 0.05$; ns, not significant.

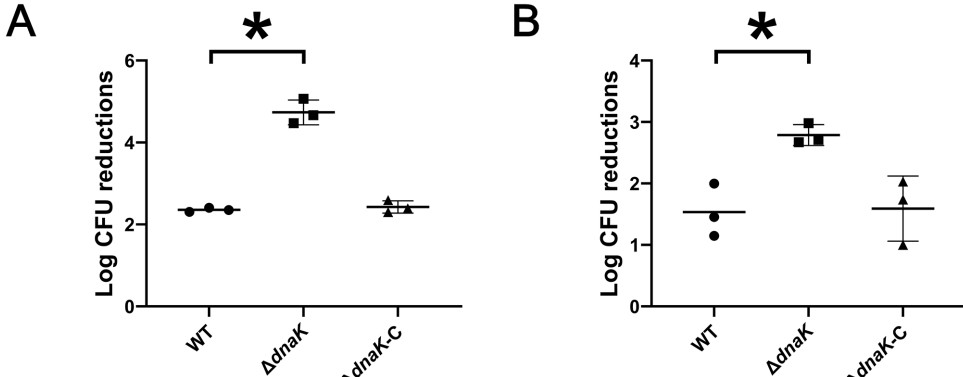

**Fig 4. Sensitivity of the *dnaK*-deficient strain to oxidative and acid stress. (A)** Survival of strains following exposure to 5 mM hydrogen peroxide for 1 h. **(B)** Survival after incubation in TSB adjusted to pH 5.5 for 2 h. Viability was quantified by CFU plating. Data are presented as means±SEM from three biological replicates. Group comparisons were evaluated using one-way ANOVA with Tukey's multiple-comparison test. $p < 0.05$.

## Discussion

The findings of this study reveal a previously uncharacterized role for DnaK in supporting the intracellular persistence and functional impact of *Staphylococcus xylosus* within breast cancer cells. While DnaK has been extensively studied in classical pathogens as a mediator of proteostasis under stress conditions [21,22], its importance in coagulase-negative staphylococci and in the context of tumor-associated microbiota has remained largely unexplored. Our results suggest that DnaK enables *S. xylosus* to withstand key microenvironmental challenges encountered within tumor cells, thereby promoting bacterial fitness and enhancing host cell resilience under mechanical stress.

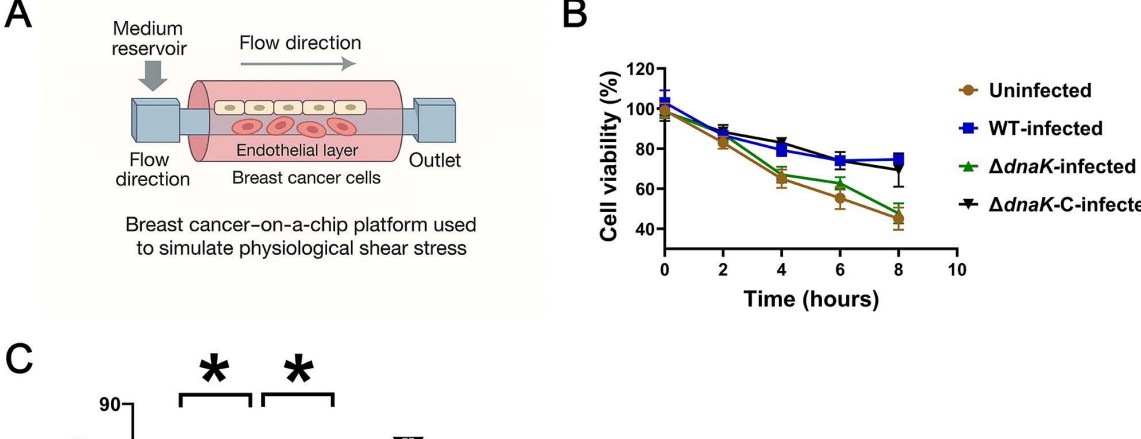

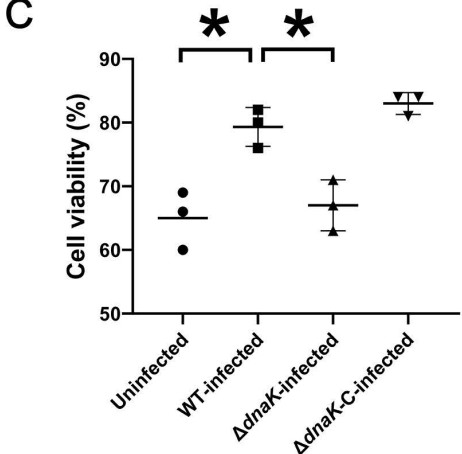

**Fig 5. Microfluidic tumor-on-a-chip model for time-resolved shear-stress exposure of S. xylosus–infected tumor cells. (A)** Diagram of the breast cancer–on–a–chip platform used to simulate physiological laminar shear stress. **(B)** Time-course analysis of tumor-cell viability under laminar shear flow. MDA-MB-231 cells were uninfected or infected with wild-type, ΔdnaK, or ΔdnaK-C Staphylococcus xylosus. Cell viability was quantified at the indicated time points using a live/dead fluorescence assay and normalized to uninfected cells at 0 h. Data represent means ± SEM from three independent microfluidic experiments. **(C)** Quantification of tumor-cell viability at 4 h of shear-stress exposure. Each data point represents an independent microfluidic experiment. Statistical significance was determined by one-way ANOVA with Tukey's post hoc test. *p < 0.05; ns, not significant.*

One of the most striking features of tumor-associated bacteria is their ability to persist intracellularly in the face of host-imposed stressors, such as oxidative bursts, low pH, hypoxia, and exposure to antimicrobial peptides [5,7]. Within this context, the stress response machinery of the bacterium becomes critically important. Our study identifies DnaK as a key molecular chaperone required for adaptation to oxidative and acid stress, but not for survival under hypoxic or antimicrobial peptide conditions. This functional specificity is consistent with the biochemical role of DnaK in stabilizing and refolding oxidatively damaged proteins [16], and supports previous findings in *Listeria monocytogenes* and *Escherichia coli*, where DnaK-deficient strains exhibit enhanced protein aggregation under ROS-generating environments [16,23,24].

Interestingly, the ΔdnaK mutant exhibited no impairment in adhesion or invasion, highlighting that DnaK's primary role in *S. xylosus* is not to mediate entry but rather to sustain intracellular viability. This observation aligns with reports in *Salmonella enterica*, where DnaK disruption does not alter initial host cell interaction but significantly reduces intracellular replication [16]. The tumor cytoplasm, particularly within breast cancer cells, is enriched in ROS, acidic metabolites, and stress granules [25–27], all of which likely create a selective environment in which only stress-adapted microbes can survive.

In addition to its direct role in bacterial stress tolerance, it is also possible that DnaK-dependent bacterial persistence indirectly influences host cell physiology. Previous studies have shown that bacterial Hsp70 family proteins can modulate host stress-response pathways, including proteins involved in DNA damage repair and apoptosis such as PARP1 and p53 [28–32]. Alterations in host cell stress responses or viability may, in turn, affect the intracellular niche available for bacterial survival. Although the present study does not directly address host-mediated mechanisms, this possibility warrants further investigation and highlights the complex, bidirectional nature of host–microbe interactions within tumor cells.

Beyond microbial survival, our study demonstrates that DnaK-dependent persistence of *S. xylosus* also benefits the host tumor cell by enhancing resistance to biomechanical stress, as modeled by a tumor-on-a-chip system. This observation resonates with recent reports showing that intratumoral bacteria can reinforce epithelial barriers [33] and promote metastatic potential by modulating cytoskeletal tension and host gene expression [11]. While the exact mechanism by which intracellular bacteria confer mechanical resilience remains to be elucidated, one plausible explanation is that persistent bacterial presence may modulate host stress pathways, reduce apoptosis, or alter actomyosin contractility [6,34]. DnaK, by ensuring bacterial survival in such hostile intracellular conditions, may therefore indirectly contribute to tumor cell plasticity and survival during intravascular shear or detachment from the extracellular matrix.

The broader implications of these findings extend to our understanding of tumor–microbiome interactions. The presence of viable, metabolically active bacteria within tumors challenges long-held assumptions about sterility in solid tumors and raises new questions about microbial contributions to tumor progression, therapy resistance, and immune modulation [35–37]. Given that DnaK is a highly conserved bacterial protein and immunodominant antigen, its role in shaping host immune responses warrants further investigation [38]. Whether DnaK-expressing *S. xylosus* within tumors contribute to immune evasion, chronic inflammation, or checkpoint blockade resistance remains an intriguing possibility that merits future study.

In conclusion, our work positions DnaK as a critical factor in the survival strategy of *S. xylosus* within breast tumors, enabling bacterial persistence and potentially enhancing host cell stress tolerance. These findings offer new insights into the adaptive strategies of intratumoral microbes and underscore the importance of bacterial proteostasis in host–microbe interactions within cancer. Targeting microbial stress response pathways such as DnaK may represent a novel avenue for modulating tumor–microbiota dynamics and improving cancer therapy outcomes.

## Conclusions

This study identifies the conserved molecular chaperone DnaK as a key determinant of *Staphylococcus xylosus* intracellular persistence within breast cancer cells. Although its canonical function in oxidative and acid stress resistance is known in other bacteria, our work demonstrates that *S. xylosus* exploits this stress-protective machinery to adapt to the hostile tumor intracellular niche. By maintaining bacterial viability under oxidative and acidic stress, DnaK enables sustained bacterial presence that in turn enhances the mechanical resilience of host tumor cells under shear stress. These findings reveal a previously unrecognized link between bacterial stress adaptation and tumor cell biomechanics, suggesting that microbial stress response systems may contribute to cancer cell survival in physically and immunologically challenging environments. Targeting such microbe–tumor interactions could open new avenues for microbiome-informed therapeutic strategies.

## Acknowledgments

Authors have no acknowledgments to declare.

## Author contributions

**Conceptualization:** Yi Cheng.

**Investigation:** Lei Ye, Guozheng Yu, Yi Cheng, Lijuan Fan.

**Methodology:** Lei Ye, Guozheng Yu, Lijuan Fan.

**Project administration:** Lei Ye, Lijuan Fan.

**Resources:** Lei Ye, Lijuan Fan.

**Software:** Lei Ye.

**Supervision:** Lijuan Fan.

**Validation:** Lei Ye, Lijuan Fan.

**Visualization:** Lijuan Fan.

**Writing – original draft:** Lei Ye, Lijuan Fan.

**Writing – review & editing:** Lei Ye, Guozheng Yu, Yi Cheng, Lijuan Fan.

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
