## [Decision Letter · Decision Letter 0]

17 Nov 2025

Dear Dr. Fan,

Thank you for submitting your manuscript to PLOS ONE. After careful consideration, we feel that it has merit but does not fully meet PLOS ONE’s publication criteria as it currently stands. Therefore, we invite you to submit a revised version of the manuscript that addresses the points raised during the review process.

We look forward to receiving your revised manuscript.

Kind regards,

Shih-Chao Lin, Ph.D.

Academic Editor

PLOS ONE

Journal Requirements:

Reviewers' comments:

Reviewer's Responses to Questions

**Comments to the Author**

1. Is the manuscript technically sound, and do the data support the conclusions?

Reviewer #1: Yes

Reviewer #2: Yes

2. Has the statistical analysis been performed appropriately and rigorously?

Reviewer #1: Yes

Reviewer #2: Yes

3. Have the authors made all data underlying the findings in their manuscript fully available?

Reviewer #1: Yes

Reviewer #2: Yes

4. Is the manuscript presented in an intelligible fashion and written in standard English?

Reviewer #1: Yes

Reviewer #2: Yes

Reviewer #1: Along with many articles published recently exploring the roles of intratumoral bacteria in modulating cancer cell physiology, PONE-D-25-55502 presents some novel findings on the importance of dnaK in stimulating the metastatic potential breast cancer cell. This article can be considered for publication after a major revision and points that need to be revised are shown in the following:

1. Improved expression of microfluidic shear stress assay shown in figure 5: Instead of presenting cell viability following a fixed time period (8 h) of laminar flow, a better understanding on the role of dnaK in modulating the metastatic potential of the breast cancer cell line MDA-MB-231 can be achieved using a kinetic presentation of cell viability. It would be good to revise figure 5 by showing how the survival rates of uninfected cancer cells and cells infected with wild-type, dnak-mutant and dnak-C strain change according to hour after laminar flow using the viability of uninfected cells at the shortest flow time as the basis for comparison as the resilience of cancer cells to shear stress should change gradually after bacterial infection.

2. Reorganization of abstract: Based on the figures shown in this submission, most readers would not be surprised to know the importance of dnak, a bacterial version of Hsp70, in counteracting acid and oxidative stresses. The most valuable data of this submission should be the ability of the dnak of Staphylococcus xylosus to enhance cancer cell viability under shear stress, therefore promoting the metastatic potential of breast cancer cells. Information on this point are poorly revealed in the abstract of the original submission and they need to be emphasized in the revised abstract while reducing the text that describe information not very surprising to the readers.

3. Title change: --mechanical resilience to breast tumor cells should change to-- mechanical resilience to a human breast cancer cell line as a human breast cancer line, not a whole tumor model, was used for studying the influence of intratumoral bacteria on tumor metastatic potential.

4. The source of wild-type Staphylococcus xylosus: It should be made clear who actually isolated this strain from human breast tumor. Was the bacterial strain provided by the authors contributing reference 11 as shown in materials and methods? or isolated by the authors of PONE-D-25-55502 themselves as shown in Table 1 because it says “this study”?

5. Which is the correct term ? DnaK or dnaK?

Reviewer #2: The manuscript presents interesting findings regarding the role of DnaK in Staphylococcus xylosus intracellular survival and stress adaptation and contribute to mechanical resistance of breast tumor cells using breast cancer cell line MDA-MB-231. The topic is relevant and potentially impactful; however, there are several concerns regarding data interpretation, methodological consistency, and clarity of presentation that should be addressed before considering the manuscript for publication.

A. Initial Bacterial Inoculum and Growth Data (Figure 1B & Figure 3A):

In Figure 1B, the authors report an initial bacterial density of approximately 1 × 10⁸ CFU/mL corresponding to an OD₆₀₀ of 0.05. The 1 × 10⁸ CFU/mL typically aligns with OD₆₀₀ values of 0.08–0.1 (equivalent to 0.5 McFarland). The reported data therefore appear inconsistent and should be clarified.

Additionally, the increase in bacterial count shown in Figure 1B reflects only about a one-log increase, which seems minimal and raises questions about the growth phase or conditions used during the experiment.

In Figure 3A, the starting OD₆₀₀ is reported to be around 0.15, equivalent to approximately 1 × 10⁸ CFU/mL which is not consistent with the inoculum in Figure 1B at OD 600 0.05. The authors should clarify why different starting concentrations were used and whether this variation could affect the observed growth trends.

B. Invasion and Intracellular Survival Assays (Figure 2):

The invasion assay results (Figure 2C) indicate that all tested strains have comparable intracellular entry levels, suggesting DnaK is not required for host cell invasion. However, Figure 2D shows that the ΔDnaK mutant exhibits significantly reduced intracellular survival approximately a threefold decrease relative to the wild type.

It is not clear how the assay differentiates between the intracellular level in the invasion assay and that for intracellular survival.

Moreover, the authors conclude that DnaK is specifically required for long-term intracellular persistence but not for host cell entry. This interpretation may be premature, given that the post infection incubation time is only 2 hours. Two hours may reflect early post-entry survival rather than long-term persistence.

C. Role of DnaK and Potential Effects on Host Cells:

The DnaK can modulate host cell proteins such as PARP1 and p53, which are essential for maintaining genomic integrity. Given this, it is probable that the presence or absence of DnaK could influence host cell viability or response, thereby indirectly affecting bacterial intracellular survival. The authors should discuss this possibility and consider whether observed differences might partly result from altered host cell physiology rather than direct bacterial adaptation.

**Do you want your identity to be public for this peer review?** For information about this choice, including consent withdrawal, please see our Privacy Policy

Reviewer #1: **Yes:** Todd Hsu

Reviewer #2: No

---

## [Author Response · Author response to Decision Letter 1]

18 Dec 2025

Response to Reviewers

Reviewer #1

Comment 1

Improved expression of microfluidic shear stress assay shown in figure 5: Instead of presenting cell viability following a fixed time period (8 h) of laminar flow, a better understanding on the role of dnaK in modulating the metastatic potential of the breast cancer cell line MDA-MB-231 can be achieved using a kinetic presentation of cell viability. It would be good to revise figure 5 by showing how the survival rates of uninfected cancer cells and cells infected with wild-type, dnak-mutant and dnak-C strain change according to hour after laminar flow using the viability of uninfected cells at the shortest flow time as the basis for comparison as the resilience of cancer cells to shear stress should change gradually after bacterial infection.

Response:

We fully agree with this insightful suggestion. To more clearly capture the temporal dynamics of tumor-cell responses to mechanical stress, we revised the microfluidic assay to include a time-course analysis of cell viability under laminar shear flow. Tumor-cell viability was quantified at multiple time points (0, 2, 4, 6, and 8 h) following flow initiation and normalized to uninfected cells at 0 h. As shown in the revised Figure 5, cells infected with wild-type or complemented Staphylococcus xylosus maintained higher viability during shear-stress exposure, whereas ΔdnaK-infected cells exhibited reduced survival comparable to uninfected controls. In addition, viability at the 4 h time point was presented separately to facilitate direct statistical comparison among groups. Corresponding revisions have been made to the Results section and the Figure 5 legend.

The revised content is as follows:

“DnaK enables S. xylosus-infected tumor cells to resist mechanical stress in a microfluidic model

To examine the impact of DnaK-mediated bacterial persistence on tumor-cell survival under mechanical stress, MDA-MB-231 cells were infected with wild-type, ΔdnaK, or ΔdnaK-C Staphylococcus xylosus and subjected to laminar shear stress using a breast cancer–on–a–chip microfluidic platform (Figure 5A). Time-course analysis revealed that cells infected with wild-type or ΔdnaK-C strains maintained higher viability during shear-stress exposure compared with uninfected cells, whereas ΔdnaK-infected cells showed reduced survival that closely resembled uninfected controls (Figure 5B). Consistent with these observations, direct comparison at the 4 h time point demonstrated significantly increased viability in cells infected with wild-type or complemented strains relative to both uninfected and ΔdnaK-infected cells (Figure 5C). Together, these data indicate that DnaK is required for S. xylosus–associated enhancement of tumor-cell survival under mechanical stress, and that this phenotype is fully restored by genetic complementation.

Figure 5. Microfluidic tumor-on-a-chip model for time-resolved shear-stress exposure of S. xylosus–infected tumor cells. (A) Diagram of the breast cancer–on–a–chip platform used to simulate physiological laminar shear stress. (B) Time-course analysis of tumor-cell viability under laminar shear flow. MDA-MB-231 cells were uninfected or infected with wild-type, ΔdnaK, or ΔdnaK-C Staphylococcus xylosus. Cell viability was quantified at the indicated time points using a live/dead fluorescence assay and normalized to uninfected cells at 0 h. Data represent means ± SEM from three independent microfluidic experiments. (C) Quantification of tumor-cell viability at 4 h of shear-stress exposure. Each data point represents an independent microfluidic experiment. Statistical significance was determined by one-way ANOVA with Tukey’s post hoc test. p < 0.05; ns, not significant.”

Comment 2

Reorganization of abstract: Based on the figures shown in this submission, most readers would not be surprised to know the importance of dnak, a bacterial version of Hsp70, in counteracting acid and oxidative stresses. The most valuable data of this submission should be the ability of the dnak of Staphylococcus xylosus to enhance cancer cell viability under shear stress, therefore promoting the metastatic potential of breast cancer cells. Information on this point are poorly revealed in the abstract of the original submission and they need to be emphasized in the revised abstract while reducing the text that describe information not very surprising to the readers.

Response:

We appreciate this recommendation and have substantially revised the Abstract to highlight the central and novel finding of this study.

The revised abstract is as follows:

“Intratumoral Staphylococcus xylosus enhances the ability of breast cancer cells to survive mechanical shear stress, a critical barrier encountered during hematogenous metastasis. However, the bacterial determinants underlying this effect remain unclear. Here, we identify the bacterial molecular chaperone DnaK as a key factor enabling S. xylosus to promote shear-stress tolerance in a human breast cancer cell line. Deletion of dnaK did not affect bacterial adhesion to or invasion of MDA-MB-231 cells but significantly reduced sustained intracellular survival. Under oxidative and acidic stress conditions, the ΔdnaK mutant showed reduced survival compared with the wild-type strain, and its ability to enhance tumor-cell viability under shear stress was markedly impaired. Using a breast cancer–on–a–chip microfluidic model, we demonstrate that infection with wild-type or complemented Staphylococcus xylosus confers increased tumor-cell viability under laminar shear stress in a time-dependent manner, whereas cells infected with the ΔdnaK mutant fail to acquire shear-stress resistance and resemble uninfected controls. Together, these findings establish DnaK-dependent intracellular persistence of S. xylosus as a critical determinant of tumor-cell survival under mechanical stress, linking a conserved bacterial stress-response protein to cancer cell biomechanics in a metastasis-relevant context.”

Comment 3

Title change: --mechanical resilience to breast tumor cells should change to-- mechanical resilience to a human breast cancer cell line as a human breast cancer line, not a whole tumor model, was used for studying the influence of intratumoral bacteria on tumor metastatic potential.

Response:

We thank the reviewer for this important clarification. We agree that the original title could be misleading regarding the experimental model used. Accordingly, the title has been revised to “DnaK Supports Intracellular Persistence of Staphylococcus xylosus and Confers Mechanical Resilience to a Human Breast Cancer Cell Line,” which accurately reflects that the mechanical stress experiments were performed using a human breast cancer cell line rather than a whole tumor model.

Comment 4

The source of wild-type Staphylococcus xylosus: It should be made clear who actually isolated this strain from human breast tumor. Was the bacterial strain provided by the authors contributing reference 11 as shown in materials and methods? or isolated by the authors of PONE-D-25-55502 themselves as shown in Table 1 because it says “this study”?

Response:

We thank the reviewer for pointing out this inconsistency. We apologize for the unclear wording in the original version of Table 1. The wild-type Staphylococcus xylosus strain used in this study was provided by the authors of reference 11 and was not isolated by our group in the present study. Table 1 has been corrected accordingly, and the source of the wild-type strain is now listed as “Provided by [11].”

Comment 5

Which is the correct term ? DnaK or dnaK?

Response:

We appreciate this comment and have standardized nomenclature throughout the manuscript: dnaK (italicized) refers to the gene, and DnaK refers to the protein. All instances in the text, figures, and tables have been corrected accordingly.

Reviewer #2

Comment A

Initial Bacterial Inoculum and Growth Data (Figure 1B & Figure 3A):

In Figure 1B, the authors report an initial bacterial density of approximately 1 × 10⁸ CFU/mL corresponding to an OD₆₀₀ of 0.05. The 1 × 10⁸ CFU/mL typically aligns with OD₆₀₀ values of 0.08–0.1 (equivalent to 0.5 McFarland). The reported data therefore appear inconsistent and should be clarified.

Additionally, the increase in bacterial count shown in Figure 1B reflects only about a one-log increase, which seems minimal and raises questions about the growth phase or conditions used during the experiment.

In Figure 3A, the starting OD₆₀₀ is reported to be around 0.15, equivalent to approximately 1 × 10⁸ CFU/mL which is not consistent with the inoculum in Figure 1B at OD 600 0.05. The authors should clarify why different starting concentrations were used and whether this variation could affect the observed growth trends.

Response:

We thank the reviewer for this careful assessment of the growth data and for pointing out potential confusion regarding the initial inoculum. We apologize for the unclear presentation in the original figures.

In both Figure 1B and Figure 3A, the actual initial optical density (OD₆₀₀) values were approximately 0.1, which empirically corresponded to ~1 × 10⁸ CFU/mL for Staphylococcus xylosus under our experimental conditions. In the original version of Figure 1B, the y-axis scale made this less apparent. To improve clarity, we have revised Figure 1B by adding appropriate axis scaling, which now more clearly reflects the initial OD₆₀₀ values.

Regarding the magnitude of growth observed in Figure 1B, the experiment was designed to compare the relative growth kinetics of wild-type, ΔdnaK, and ΔdnaK-C strains during early exponential growth rather than to capture maximal cell density. The observation window was therefore intentionally limited, resulting in an approximately one-log increase in CFU that was sufficient to assess whether deletion of dnaK affected in vitro growth.

For Figure 3A, the same initial target OD₆₀₀ (~0.1) and corresponding CFU (~1 × 10⁸ CFU/mL) were used. Differences in visual appearance between Figures 1B and 3A arise from distinct y-axis scaling rather than differences in starting inoculum.

The revised Figure 1 is as follows:

The revised Figure 3 is as follows:

Comment B

Invasion and Intracellular Survival Assays (Figure 2):

The invasion assay results (Figure 2C) indicate that all tested strains have comparable intracellular entry levels, suggesting DnaK is not required for host cell invasion. However, Figure 2D shows that the ΔDnaK mutant exhibits significantly reduced intracellular survival approximately a threefold decrease relative to the wild type.

It is not clear how the assay differentiates between the intracellular level in the invasion assay and that for intracellular survival.

Moreover, the authors conclude that DnaK is specifically required for long-term intracellular persistence but not for host cell entry. This interpretation may be premature, given that the post infection incubation time is only 2 hours. Two hours may reflect early post-entry survival rather than long-term persistence.

Response:

We thank the reviewer for raising this important point regarding assay distinction and terminology. We would like to clarify that invasion and intracellular persistence were assessed at different post-infection time points using distinct experimental endpoints. Adhesion and invasion were quantified at 2 h post-infection, immediately following gentamicin treatment, to assess bacterial entry into host cells. In contrast, intracellular survival was evaluated at 24 h post-infection by enumerating viable intracellular CFUs following extended gentamicin protection, reflecting long-term intracellular persistence rather than early post-entry survival. To avoid any potential confusion, we have revised the Results section to explicitly indicate the corresponding time points and to clearly distinguish initial invasion from long-term intracellular persistence.

The revised Results section is as follows:

“DnaK is essential for intracellular persistence but not for adhesion or invasion

The ability of wild-type, ΔdnaK, and ΔdnaK-C strains to interact with breast cancer cells was assessed using an in vitro infection model. Bacterial adhesion and invasion were quantified at 2 h post-infection, and all strains exhibited comparable levels of host-cell association and intracellular entry at this early time point (Figure 2A, 2B).To distinguish initial entry from intracellular persistence, total bacterial burden per well was next quantified at 24 h post-infection using S. xylosus–specific qPCR, which revealed no significant differences among the three strains, indicating equivalent initial infection levels and bacterial association with host cells (Figure 2C). In contrast, intracellular survival was assessed at 24 h post-infection following gentamicin protection and host-cell lysis, and enumeration of viable intracellular CFUs demonstrated that the ΔdnaK mutant exhibited a more than three-fold reduction in intracellular survival compared with the wild-type strain (Figure 2D). This defect was fully restored in the complemented strain. Together, these results demonstrate that DnaK is required for long-term intracellular persistence at later post-infection time points, but is dispensable for host-cell entry at early stages of infection.”

Comment C

Role of DnaK and Potential Effects on Host Cells:

The DnaK can modulate host cell proteins such as PARP1 and p53, which are essential for maintaining genomic integrity. Given this, it is probable that the presence or absence of DnaK could influence host cell viability or response, thereby indirectly affecting bacterial intracellular survival. The authors should discuss this possibility and consider whether observed differences might partly result from altered host cell physiology rather than direct bacterial adaptation.

Response:

We thank the reviewer for this insightful comment. We agree that, in addition to its role in bacterial stress tolerance, DnaK-dependent bacterial persistence may indirectly influence host cell physiology. We have revised the Discussion to acknowledge that bacterial Hsp70 family proteins, including DnaK, have been reported to modulate host stress-response pathways such as those involving PARP1 and p53, which could in turn affect the intracellular niche available for bacterial survival. While dissecting host-mediated mechanisms was beyond the scope of the current study, this possibility is now explicitly discussed as an important direction for future investigation.

The added Discussion section is as follows:

“In addition to its direct role in bacterial stress tolerance, it is also possible that DnaK-dependent bacterial persistence indirectly influences host cell physiology. Previous studies have shown that bacterial Hsp70 family proteins can modulate host stress-response pathways, including proteins involved in DNA damage repair and apoptosis such as PARP1 and p53[28-32]. Alterations in host cell stress responses or viability may, in turn, affect the intracellular niche available for bacterial survival. Although the present study does not directly address host-mediated mechanisms, this possibility warrants further investigation and highlights the complex, bidirectional nature of host–microbe interactions within tumor cells.”

---

## [Decision Letter · Decision Letter 1]

2 Jan 2026

DnaK Supports Intracellular Persistence of Staphylococcus xylosus and Confers Mechanical Resilience to a Human Breast Cancer Cell Line

PONE-D-25-55502R1

Dear Dr. Fan,

We’re pleased to inform you that your manuscript has been judged scientifically suitable for publication and will be formally accepted for publication once it meets all outstanding technical requirements.

Kind regards,

Shih-Chao Lin, Ph.D.

Academic Editor

PLOS One

Additional Editor Comments (optional):

Reviewers' comments:

Reviewer's Responses to Questions

**Comments to the Author**

Reviewer #1: All comments have been addressed

Reviewer #2: All comments have been addressed

2. Is the manuscript technically sound, and do the data support the conclusions?

Reviewer #1: Yes

Reviewer #2: Yes

3. Has the statistical analysis been performed appropriately and rigorously?

Reviewer #1: Yes

Reviewer #2: I Don't Know

4. Have the authors made all data underlying the findings in their manuscript fully available?

Reviewer #1: Yes

Reviewer #2: Yes

5. Is the manuscript presented in an intelligible fashion and written in standard English?

Reviewer #1: Yes

Reviewer #2: Yes

Reviewer #1: All necessary revisions of title, source of bacteria, and most importantly a time-course monitoring of the viability of cancer cells infected by wild-type and mutant Staphylococcus xylosus against mechanical shear force have been made. Hence, PONE-D-25-55502R1 can be accepted for publication.

Reviewer #2: The authors have revised the manuscript and made the required amendments and have adequately addressed all raised comments

**Do you want your identity to be public for this peer review?** For information about this choice, including consent withdrawal, please see our Privacy Policy

Reviewer #1: **Yes:** Todd Hsu

Reviewer #2: No

---

## [Editor Report · Acceptance letter]

PONE-D-25-55502R1

PLOS One

Dear Dr. Fan,

I'm pleased to inform you that your manuscript has been deemed suitable for publication in PLOS One. Congratulations! Your manuscript is now being handed over to our production team.

Kind regards,

on behalf of

Dr. Shih-Chao Lin

Academic Editor

PLOS One